# Dynamic Mode Decomposition of Fluorescence Loss in Photobleaching Microscopy Data for Model-Free Analysis of Protein Transport and Aggregation in Living Cells

**DOI:** 10.3390/s22134731

**Published:** 2022-06-23

**Authors:** Daniel Wüstner

**Affiliations:** Department of Biochemistry and Molecular Biology and Physics of Life Sciences (PhyLife) Center, University of Southern Denmark, Campusvej 55, DK-5230 Odense, Denmark; wuestner@bmb.sdu.dk

**Keywords:** photobleaching, time-lapse microscopy, model reduction, biomolecular condensates, protein aggregation, intracellular transport

## Abstract

The phase separation and aggregation of proteins are hallmarks of many neurodegenerative diseases. These processes can be studied in living cells using fluorescent protein constructs and quantitative live-cell imaging techniques, such as fluorescence recovery after photobleaching (FRAP) or the related fluorescence loss in photobleaching (FLIP). While the acquisition of FLIP images is straightforward on most commercial confocal microscope systems, the analysis and computational modeling of such data is challenging. Here, a novel model-free method is presented, which resolves complex spatiotemporal fluorescence-loss kinetics based on dynamic-mode decomposition (DMD) of FLIP live-cell image sequences. It is shown that the DMD of synthetic and experimental FLIP image series (DMD-FLIP) allows for the unequivocal discrimination of subcellular compartments, such as nuclei, cytoplasm, and protein condensates based on their differing transport and therefore fluorescence loss kinetics. By decomposing fluorescence-loss kinetics into distinct dynamic modes, DMD-FLIP will enable researchers to study protein dynamics at each time scale individually. Furthermore, it is shown that DMD-FLIP is very efficient in denoising confocal time series data. Thus, DMD-FLIP is an easy-to-use method for the model-free detection of barriers to protein diffusion, of phase-separated protein assemblies, and of insoluble protein aggregates. It should, therefore, find wide application in the analysis of protein transport and aggregation, in particular in relation to neurodegenerative diseases and the formation of protein condensates in living cells.

## 1. Introduction

Fluorescence microscopy has enormous potential in determining the dynamics and interactions of proteins in living cells. In particular, genetically encoded fluorescent protein tags, such as green fluorescent protein and its color variants, combined with powerful live-cell imaging, have revolutionized our understanding of protein dynamics and compartmentalization. The analysis of such complex imaging data is challenging, and a variety of quantitative image analysis techniques have been developed for interrogating the intracellular transport of proteins and other biomolecules. One group of such techniques comprised of fluorescence recovery after photobleaching (FRAP), fluorescence loss in photobleaching (FLIP), and continuous photobleaching (CP), employ the photobleaching of a small fraction of the tagged protein population using short laser pulses to perturb the steady-state fluorescence distribution [1,2,3]. Here, the analysis is based on determining the response of the system to the perturbation. The same principle underlies the analysis of photoactivation microscopy data [4,5]. FRAP and FLIP are easy to implement in a cell biological laboratory, requiring only conventional fluorescent protein constructs, often providing fast qualitative insight into protein dynamics. Determining the diffusion and binding constants as well as the exchange parameters between compartments from FRAP or FLIP data is also possible, but this requires elaborate computational approaches to solve and parametrize the underlying partial differential equation systems [6,7]. Often, parameter estimation is based on various assumptions, which are highly context-dependent and therefore difficult to transfer between different laboratories and setups. For selected biological systems, such as transcription-factor binding to DNA, cross-validation has shown that parameter estimates for diffusion and binding can be compared for three approaches, i.e., fluctuation, photobleaching, and single-molecule microscopy [8,9,10].

FLIP microscopy is based on the repeated photobleaching of a small cellular region and observing the fluorescence loss kinetics, caused by transport to the bleached area, throughout the entire cell. FLIP, therefore, has an advantage over FRAP; it provides not only full spatiotemporal protein dynamics, but also the sampled time scale that can be tuned by adjusting the pause time between laser pulses. Fast, repeated bleaching can result in diffusion-limited fluorescence loss with pronounced fluorescence gradients of the tagged protein, as protein diffusion is too slow to replenish bleached molecules immediately [11]. Slow bleaching, i.e., longer pauses between the laser pulses allows for diffusion to keep up with bleaching, such that diffusion is no longer limiting fluorescence-loss kinetics (bleach-limited FLIP). In this case, fluorescence-loss kinetics report primarily on barriers to diffusion, such as transport across the nuclear membrane and binding and the release of a protein to a subcellular structure or on diffusion barriers, such as that between the soma and axon in neurons [11,12]. Thus, FLIP has the potential to reveal barriers to diffusion in addition to the determination of diffusion and binding constants. In this regard, FLIP microscopy can be compared with pair correlation analysis of protein diffusion and confinement [11,13,14,15,16,17]. FLIP is also extensively used to study biomolecular condensates, which contribute to cellular compartmentalization by forming membrane-less organelles via liquid–liquid phase separation [18].

Protein aggregation is a hallmark of a number of neurodegenerative diseases, including Alzheimer’s (aggregation of Tau and Amyloid-beta protein), amyotrophic lateral sclerosis (TDP-43 protein), frontotemporal lobar degeneration (FUS protein), Parkinson’s disease (Alpha-synuclein aggregation), and Huntington’s disease (aggregation of mutant Huntingtin protein) [19]. In Huntington’s diseases and various forms of Ataxia, an extended polyglutamine stretch in disease variants of the affected protein causes a higher tendency for self-interaction via these disordered protein regions, leading to the formation of protein condensates/aggregates, sometimes called inclusion bodies [20,21,22,23]. For many of the abovementioned proteins, it has been shown that the formation of stable intracellular aggregates is preceded by the formation of soluble protein condensates, which can still exchange molecules with the surrounding cytoplasm on a time scale of minutes or less, as shown in FRAP and FLIP experiments [20,21,22,23,24,25,26]. They are also often rounded in shape in contrast to stable aggregates, which form fibrils and are of irregular shape, suggesting that, in liquid protein condensates, surface tension is a determining factor in their stability [19,23,26]. Over time, protein molecules in these droplet-like aggregates strengthen their interactions, leading to a gel-like state and, eventually, to amyloid-like fibrils, as observed in brain dissections of affected patients [19,27]. The different physico-chemical properties of soluble protein condensates versus stable fibrils are reflected in their dynamics, and, for several neurotoxic proteins, fast and slowly exchanging populations have been detected [25,28,29]. The aggregation process can be sped up by extending the polyglutamine stretch and reduced by the presence of chaperonins. For example, extending the polyglutamine tract in mutant Huntingtin (mtHtt) causes a transition from soluble liquid-like protein condensates to solid-like fibrils, as studies by FRAP, fluorescence lifetime and anisotropy imaging, single molecule tracking, and fluorescence fluctuation techniques [20,29,30,31,32,33,34,35,36,37,38]. We have previously used computational FLIP to characterize the dynamic properties of mtHtt with 73 repetitions of glutamine tagged with enhanced green fluorescent protein (eGFP-Q73). A residence time for this protein of 16–83 sec in cytoplasmic inclusion bodies was found [11,39]. Using a reaction–diffusion model, we also determined diffusion constants and rate constants for nucleo-cytoplasmic exchange of this protein, and we found that it can diffuse about 30 µm upon release from a protein condensate before being recaptured and incorporated into another aggregate [39,40]. Diffusion, rather than active transport, has been suggested to drive the aggregation of polyglutamine proteins in living cells [31].

Data-driven machine learning methods are increasingly used to analyze complex spatiotemporal processes. Dynamic mode decomposition (DMD) is a novel, data-driven approach to extract dynamic information from large spatiotemporal datasets, such as images. Increasingly, it is being applied in computer vision and biomedical imaging [41,42]. DMD has been used to detect video shots or discriminate foreground from background in videos of natural scenes; it has also been used to segment images of human kidneys and detect functional brain states by magnetic resonance imaging image sequences [43,44,45,46]. In a recent study, the potential of DMD for use in the analysis of microscopy data was demonstrated by determining the photobleaching characteristics of fluorescent probes to distinguish probe fluorescence from cellular autofluorescence [47]. Here, DMD is applied to analyze simulated FLIP images and FLIP sequences of enhanced green fluorescent protein (eGFP) and of eGFP-tagged mtHtt (eGFP-mtHtt). DMD is demonstrated to be a powerful tool to decompose the intracellular dynamics of these proteins in distinct subcellular compartments, effectively detecting barriers to intracellular protein diffusion. Using DMD, the dynamics of eGFP-mtHtt in rapidly and slowly exchanging protein aggregates is analyzed. DMD-FLIP is shown not only to allow for the kinetic identification of phase-separated proteins and stable protein aggregates, but also for the efficient spatial and temporal denoising of FLIP image sequences.

## 2. Materials and Methods

### 2.1. Cell Culture and Transfection

McArdle RH7777 (McA) cells expressing the enhanced green fluorescent protein (eGFP) were reported and used for FLIP experiments, as described previously [11]. They were grown in DMEM with 4.5 g/L glucose, supplemented with 10% heat-inactivated FCS and antibiotics. Chinese hamster ovarian (CHO) cells were purchased from ATCC (www.atcc.org; LGC Standards Office Europe, AB, Boras, Sweden) and grown in bicarbonate-buffered Ham’s F-12 medium supplemented with 5% fetal calf serum (FCS) and antibiotics. McA and CHO cells were routinely passaged in plastic tissue culture dishes. CHO cells were transiently transfected with mHtt bearing the N-terminus of Huntingtin and an extended polyglutamine stretch of 145 residues and a C-terminal eGFP-tag, using lipofectamine, following the manufacturer’s instructions (SIGMA-Aldrich, Søborg, Denmark). All cells were seeded on microscope slide dishes coated with poly-D-lysine, 2–3 days prior to experiments and kept in buffer medium containing 150 mM NaCl, 5 mM KCl, 1 mM CaCl_2_, 1 mM MgCl_2_, 5 mM glucose, and 20 mM HEPES (pH 7.4) for imaging.

### 2.2. Image Simulation

To validate the procedure, synthetic FLIP image sequences were generated using the Macro language of ImageJ (https://imagej.nih.gov/ij/ accessed on 6 February 2021), as described previously [11]. This included a three-compartment model for the protein exchange between nucleus, *N*, and cytoplasm, *C*, with rate constants, *k*_1_ and *k*_−1_; the release of protein from cytoplasmic aggregates, *A*, with rate constant, *k*_2_; as well as photobleaching in a localized region of the cytoplasm with rate constant, *k*_3_. Assuming a well-mixed system (i.e., bleach-limited FLIP, see Introduction), diffusion can be ignored, and one obtains the analytical solutions given in Appendix B.

### 2.3. FLIP Microscopy of Fluorescent Proteins in Living Mammalian Cells

In FLIP microscopy, a strong and short laser pulse is used to photobleach a suitably tagged biomolecule (e.g., a labeled protein, lipid, or RNA molecule) locally, and the fluorescence loss of the tagged molecule in other cellular regions is monitored (Figure 1A). If the labeled molecular species, for example, eGFP-mtHtt, can exchange between its initial position and the bleached region, the fluorescence loss will be observed in the area of its initial localization. If the protein is confined to a region, e.g., to subcellular aggregates, no fluorescence loss will be observed in those aggregates (green circular regions in Figure 1A). If some protein exchange can take place, for example between the cytoplasm and the phase-separated, but liquid-like, protein condensates (green ellipsoidal regions in Figure 1A), it will be delayed compared to the free cytosolic protein pool. Thus, the fluorescence-loss kinetics report about the exchange dynamics of a tagged protein translocating between subcellular compartments. FLIP microscopy was performed using a confocal laser-scanning fluorescence microscope (Zeiss LSM 510 META, Zeiss, Jena, Germany) equipped with a 63x 1.4 NA plan Apochromat water-immersion objective with temperature control at 37 °C (Zeiss, Jena, Germany). The fluorescence of eGFP was collected with a 505–530 bandpass filter after excitation with a 25-milliwatt argon laser emitting at 488 nm. FLIP experiments were performed by first defining regions of interest which were repeatedly bleached, while an image was acquired with reduced laser power (0.5% output) at the start of the experiment and after each bleaching event. A pause between the bleaching events ensured some recovery in the bleached region. Images were acquired using the time-lapse function of the Zeiss LSM510 Meta confocal system. The microscope was located at a nitrogen-floated table to prevent vibrations and focus drift and contained a temperature-controlled stage maintained at 35 ± 1 °C.

### 2.4. Outline of the DMD Method Applied to Fluorescence Microscopy Data

The first step in applying DMD to FLIP microscopy data is that the FLIP image stack, *I*(*x*, *y*, *t*), of the fluorescence intensity, *I*, as a function of spatial coordinates, *x*, *y* and time, *t*, is reshaped into *u*∙*v*∙1 column vectors, $\bar{x_{n}}$, called snapshots in the DMD literature, which become arranged in a *j* times *k* data matrix, X (Figure 1B). This matrix consists of *i* rows, which are the pixels of each image reshaped into a vector, and *k* columns, each representing one time point of the original image sequence [41,45]:(1)$$X=\left[\bar{x_{1}},\;\bar{x_{2}},\;\cdots ,\bar{x_{n}},\ldots \bar{x_{k-1}}\right]$$

Here, the index *n* = 1, …, *k* − 1 indicates the frame number that is equivalent to the time axis of the video image sequence. Separately, a second matrix is formed from the video sequence in which the second to *k*th frame is reshaped as described above giving (Figure 1B, most right matrix):(2)$$X^{\prime }=\left[\;\bar{x_{2}},\;\cdots ,\bar{x_{n}},\ldots \bar{x_{k}}\right]$$

DMD aims to find the optimal mapping between discrete time steps, Δ*t*, represented by these matrices, which describes the advancement of the system in the time described by the matrix *A*, which approximates the Koopman operator [42]:
*X*’ = *A*∙*X*
(3)



This can be achieved as outlined in Figure 1C, by minimizing the Frobenius norm, ${\left\|\cdot \right\|}_{F}$ [42]:(4)$$A∶=argmin\|X^{\prime }-A\cdot X{\|}_{F}=X^{\prime }\cdot X^{*}$$

One can find the pseudoinverse of the first data matrix, $X^{*}$, by using a singular value decomposition (SVD) of *X* into unitary matrices *U* and *V** with singular values in the diagonal matrix Σ:
*X* = *U*∙Σ∙*V**
(5)



The original matrices *X* and *X′* are of size *i* × *k*, with many more rows (i.e., pixels, typically 512 × 512, resulting in 262,144 rows, *i*) than time points (i.e., columns, *k*). Thus, *A* is a ‘tall-skinny’ matrix, since image dimension is larger than number of snapshots in time (Figure 1B). Accordingly, *A* has at most, *k* nonzero singular values and corresponding singular vectors and is maximally of rank *k* [42]. One can therefore approximate *A* by calculating its projection onto the leading singular vectors, which gives a much smaller matrix *A*′ of maximal size *k* × *k*. We obtain [42,43]:
*A*’= *U*’*∙*A*∙*U*’ = *U*’*∙*X*_2_∙*V’∙Σ’*^−1^
(6)


Often fewer, i.e., only *r* singular values are needed to describe the signal in the data matrix *A* (i.e., r << *k*), and one can employ a thresholding method to discard all singular values larger than *r*. Accordingly, the signal content of the full matrix is retained, while the noise is efficiently removed [48,49]. The optimal threshold for the truncation of singular values can be found automatically using the method of Gavish and Donoho [49]. Subsequently, a spectral decomposition of *A*’ provides the DMD modes as eigenfunctions *φ_j_*, (i.e., eigenvectors of *A*, also called DMD modes), corresponding eigenvalues, *λ_j_* (so-called DMD eigenvalues), and mode amplitudes *b_j_* according to:(7)$$x_{r}=\overset{r}{\underset{j=1}{\sum }}\varphi_{j}\cdot \lambda_{j}^{k-1}\cdot b_{j}$$

The eigenvalues determined for a rank-*r* decomposition of the system matrix *A*, *λ_j_*, for *j* = 1, …, *r*, are logarithmically scaled and divided by the interval time (i.e., the acquisition time in the case of bleach stacks, Δ*t*): (8)$$\omega_{j}=log\left(\lambda_{j}\right)/\Delta t$$

For a continuous-time system using Equation (8), this becomes:(9)$$x\left(t\right)=\overset{r}{\underset{j=1}{\sum }}\varphi_{j}\cdot e^{\omega_{j}\cdot t}\cdot b_{j}$$

Thus, the entire FLIP image series is approximated as the sum of eigenfunctions, their weights, and respective eigenvalues (Figure 1). DMD and its accompanying analysis were carried out in Python using Jupyter notebooks (https://jupyter.org/ accessed on 4 August 2020) and PyDMD, a python library for DMD calculations [50]. FLIP simulations were implemented as Macros in ImageJ (ImageJ (nih.gov)), as described previously [11]. To account for the diffusion of aggregates, a Monte Carlo simulation was used, in which two random numbers were generated from a uniform distribution and transformed to a Gaussian distribution using the Box–Muller method in an ImageJ macro [51]. For a constant diffusion coefficient and fixed time step, a Gaussian step-length distribution is known to exactly simulate normal diffusion [52]. For a simulated pixel size of 0.125 µm and one frame per second, this corresponds to the slow diffusion of the aggregates with a diffusion constant of D = 0.004 µm^2^/s, which is comparable to experimental findings for diffusion of small eGFP-mtHtt aggregates in cells [11]. In simulations with flow, a constant value of five percent of a pixel was subtracted in the negative x-direction, corresponding to a flow speed of v = 0.00625 µm. All calculations were run on either a zBook laptop computer from HP with Intel i7-8850H 2.6 GHz CPU with 32 GB RAM and an M1000M NVIDIA graphics card (NVIDIA Santa Clara, CA, USA) or on a Desktop workstation with Intel i9 11900 8-core CPU with max. 5.2 GHz and a GeForce RTC 3090 graphics card from NVIDIA (NVIDIA Santa Clara, CA, USA).

## 3. Results

### 3.1. DMD of Simulated FLIP Image Sequences

The ability of DMD to decompose FLIP dynamics was first assessed on synthetic image sequences, in which bleaching was assigned to the cytoplasm and the bi-directional exchange between the cytoplasm and the nucleus was explicitly considered. The FLIP simulation also includes protein aggregates in the cytoplasm, from which the protein can be slowly released, whereby its dynamics are coupled to the fluorescence loss in the cytoplasm and nucleus (Figure 2A,B). Simulating this model requires solving a coupled system of ordinary differential equations (ODEs), as described in Appendix B. To simplify the model, protein aggregates are assumed to have already been formed at the beginning of the simulation, without explicitly considering their assembly from protein in the cytoplasm. That means that only a slow protein release from already formed aggregates is considered, with no rebinding of released protein to the aggregates (Figure 2A). This considerably simplifies the model compared to the reversible exchange between inclusion bodies and cytoplasm and is justified by the high probability of the released protein being bleached by the FLIP laser in the cytoplasm, before eventually rebinding to an aggregate [11]. By calculating the DMD of the data using the optimal-rank option in PyDMD [50], three modes were recovered, the dynamics of which are shown in Figure 2A. Mode 0 had an eigenvalue of close to 1 with slowly changing dynamics (*λ*_0_ = 0.996, Figure 2B; yellow line and Figure 2C). After rescaling to real time using Equation (8), this corresponds to an eigenvalue of *ω*_0_ = −0.00375, which closely resembles the value used for the negative of the rate constant of fluorescence loss from aggregates in the simulation (i.e., the first eigenvalue of the analytical model, see Equation (A7), *l*_1_ = −k_2_ = −0.005 s^−1^). Accordingly, this mode describes the fluorescence-loss dynamics in the aggregates and is further supported by the 2D map of mode 0 (Figure 2D). This dynamic mode has the greatest number of negative values at the site of the aggregates, and those values are multiplied with a negative-mode amplitude for mode reconstruction (Figure 2B; yellow line). This results in positive values in the mode decay stack, calculated according to Equation (9) for j = 0 (Figure 2E, ‘Mode 0’). Modes 1 and 2 describe the faster fluorescence loss in the nucleus and cytoplasm, while both modes have values close to zero for the aggregates (Figure 2B,D). This is confirmed for the reconstructed mode decays, calculated according to Equation (9) for j = 1, 2 (Figure 2E, middle and lower panels). In particular, one sees that the sum of modes 1 and 2 exactly reconstructs the nucleocytoplasmic exchange dynamics (Figure 2E lowest panel). The eigenvalues of mode 1 or 2 of the DMD (i.e., *ω*_1_ = −0.0189 and *ω*_2_ = −0.1162) are different from the eigenvalues describing the fluorescence-loss kinetics in the nucleus and cytoplasm in the FLIP simulations of Figure 2 (*l*_2_ = −0.6342 and *l*_3_ = −0.01577). The eigenvalues used in the simulations are comprised of a combination of all rate constants (*l*_2_ and *l*_3_ in the Appendix B, Equations (A8) and (A9)), and their fractional amplitude in the simulation corresponds to 4.9% and 78.1% for the second and third component, respectively. Thus, the first and third eigenfunctions comprise more than 95% of all decay in the simulation, with eigenvalues in close agreement with the DMD of the simulated FLIP series (i.e., *ω*_0_ ≈ *l*_1_ and *ω*_2_ ≈ *l*_3_, see Table 1). Similar observations were made for simulations with inert aggregates (i.e., no exchange with *l*_1_ = −k_2_ ~−0.00 s^−1^). Thus, DMD identifies the dominant modes but eventually fails to identify small-amplitude contributions correctly. Moreover, the DMD reconstruction of additional simulations showed that the different combinations of rate constants cannot be uniquely assigned to the eigenvalues obtained by DMD. Only the first eigenvalue can be determined unequivocally in all cases, as this eigenvalue describes the release from the aggregates as a single-rate constant (see Equations (A5) and (A7)). The other two eigenvalues are a combination of rate constants (see Equations (A4) and (A6)), and it is well known that the non-orthogonality of exponential functions with real eigenvalues makes unequivocal identification of particular kinetic contributions difficult in such cases [53]. It should also be noted that our previous method of analyzing FLIP image sequences using pixel-wise fitting of exponential decay functions cannot unequivocally determine all eigenvalues of the simulated FLIP data either [11]. Fitting a bi-exponential decay model implemented in PixBleach to the FLIP simulation gave poor-fitting results (not shown). 

Using a stretched exponential model, as in our previous study [11], only the first eigenvalue describing the release from the aggregates could be recovered precisely (Table 1).

It has to be emphasized, though, that DMD in its current implementation is not a fitting method, and the derived basis function does not necessarily represent the underlying biophysical mechanism. Possible extensions of the DMD method in future applications are considered in the Discussion section. Importantly, the reconstructed FLIP simulation calculated as a sum of all three modes resembles the original simulated image data very closely (Figure 3 and Appendix A). This can be inferred from the low and homogeneous square error between the simulation and the reconstruction (Figure 3A, right panels) as well as from the similar total-intensity decay (Figure 3B). It also becomes obvious that DMD efficiently denoises the FLIP image sequence, which is particularly important for later time points, when the majority of the signal is already removed (Figure 3A, lower panels). Note that the imaginary part of all modes was zero, as no intensity oscillations or lateral movement was considered in the FLIP simulation (not shown). In real experiments, aggregates might move slightly in the cells during a FLIP experiment, so the impact of the diffusion and flow of the aggregates on the DMD reconstruction quality was assessed as well. First, the random displacement of the smallest aggregate was included in the FLIP simulation. The Brownian motion of the aggregate could not be reconstructed by a low-rank DMD, which resulted in the blurring of the particle position (see Appendix C). Secondly, a directed movement component was added to the FLIP simulation to emulate diffusion with the flow of the smallest aggregate (Appendix C and Appendix A). In that case, the flow component without random motion was partially reconstructed by the DMD. Importantly, the fluorescence loss in the moving aggregates was still adequately described in both cases, showing that the decomposition of fluorescence-loss kinetics in the subcellular structure can be decoupled from the lateral displacement of those components.

### 3.2. DMD of Experimental FLIP Image Sequences of Nucleo-Cytoplasmic Exchange of eGFP

To assess the potential of DMD in the analysis of experimental FLIP image data, a FLIP experiment was carried out on McA cells expressing eGFP (see Materials and Methods). Repeated photobleaching of a small, circular regions in the cytoplasm caused the rapid fluorescence loss of eGFP. The fluorescence loss was first seen in the cytoplasm and, with a delay, also in the nucleus (Figure 4A). DMD of this data in PyDMD, using the optimal rank option, gave a rank-6 approximation of the data matrix, resulting in a very good reconstruction and denoising of the data (Figure 4A,B and Appendix A). Of note, the FLIP image stack reconstructed from DMD was significantly less noisy than the original data.

The SVD of the space–time data matrix, corresponding to the entire FLIP image stack, revealed four dominant singular values that, together, contributed almost 90% of the data variance (Appendix D). Additionally, there was a slowly decaying contribution of more than 20 singular values, but the optimal rank was set to rank 6 using the hard-threshold algorithm of Gavish and Donoho (2014) [49]. The first two modes had amplitudes close to zero and were almost constant in time (Figure 5A) with no visible structure in the 2D-mode plots (not shown). The third mode could be assigned to the neighboring cell, which did not experience any fluorescence loss and, consequently, had a constant mode amplitude with an eigenvalue close to one (*λ*_0_ = 0.999), corresponding to an eigenvalue of *ω*_0_ = 0.00 after rescaling to real time using Equation (8) (Figure 5A, green line, Figure 5B,C). Modes 3 to 5 described the fluorescence loss in the cytoplasm and nucleus with slowly decaying dynamics (Figure 5B green dashed line and red lines and Figure 5D–F). The small lateral displacements of the cell during the FLIP experiment are reflected in the non-zero imaginary contributions to modes 4 and 5 (Figure 5E,F). The reconstruction of the individual modes for this sequence, according to Equation (9), above, confirms that mode 2 accounts for the fluorescence of the unbleached neighboring cell, while modes 3 and 4 describe the fluorescence loss in the nucleus and cytoplasm (Appendix D). The results in Figure 4 and Figure 5 demonstrate that DMD of live-cell FLIP image sequences allows for the efficient image denoising and the correct description of the pixel-wise fluorescence-loss dynamics in both compartments of bleached cells but also of fluorescence in the unbleached neighboring cells.

### 3.3. DMD of Experimental FLIP Image Sequences of Soluble and Aggregated eGFP-mtHtt

To assess the ability of DMD-FLIP to discern the dynamic components in FLIP experiments with protein aggregates, eGFP-tagged mtHtt, with a 145-residue long glutamine region (eGFP-Q145), was expressed in CHO cells. The FLIP microscopy of those cells revealed the existence of stable aggregates in which the protein exchange was significantly slowed (Figure 6A, upper panel). Faster-intensity decay was found in small, mobile aggregates. Fluorescence loss is only partial in the cell shown in Figure 6, as some out-of-focus drift made longer acquisitions unreliable. Still, DMD of this FLIP sequence resulted in a very good reconstruction with a close coincidence of the integrated intensity and significantly reduced noise levels (Figure 6A, lower panel and Figure 6B). The denoising capacity of DMD-FLIP is significant, as subcellular features are more discernable in the reconstructed stack than in the raw image sequence (6C–E). Denoising by DMD-FLIP is also efficient over time, where fluorescence-loss kinetics were much smoother in the reconstructed stack than in the original FLIP image data (Figure 6F,G). The temporal intensity profiles coincide well in regions with exponential fluorescence loss (Figure 6C,D, Box 1 to 3 and Figure 6F). Even the small protein aggregate highlighted in Box 1 showed significant fluorescence loss, which was adequately described by the DMD-reconstructed FLIP stack, with significant temporal denoising compared to the original FLIP data (compare the red and blue straight lines in Figure 6F, corresponding to Box 1 in Figure 6C,D). Only in regions of the large solid-like aggregate, where the intensity decay was minimal, DMD’s intensity profile deviated somewhat from the experimental one (Figure 6C,D Box 4 and Figure 6G, dashed lines). This is likely due to the slow exchange and diffusion of proteins residing in the stable aggregates. The resulting non-exponential fluorescence loss is not adequately described by DMD. The extremely inefficient diffusion of eGFP-Q145 inside the aggregates was confirmed in separate FLIP experiments (Appendix E). The potential of DMD-FLIP to denoise FLIP image series is further demonstrated by comparison to other denoising algorithms, such as anisotropic filtering and PURE-LET denoising [54,55]. This analysis shows that DMD-FLIP is on par with (PURE-LET denoising) or even superior to (Anisotropic filtering) those methods (Appendix F). While PURE-LET denoising results in slightly higher PSNR than DMD-FLIP, the latter has a much better ability to denoise the time decay and to preserve fine structural details (Appendix F and see Appendix A). On the other hand, DMD-FLIP fails to accurately describe the sudden lateral displacement of small aggregates in the cell.

This is in line with the observations made in FLIP simulations with Brownian motion (see Appendix C and Appendix F). The optimal rank determined using the hard-thresholding method in PyDMD was rank = 2, since only two dominant singular values were found (Figure 7A). Thus, only two modes were considered in further analysis and used for image reconstruction. While mode 0 describes the slowly decaying fluorescence of eGFP-Q145 in cytoplasm and in the aggregates (*ω*_0_ = −0.0051 s^−1^), mode 1 comprises the faster decaying, likely monomeric or oligomeric, mobile eGFP-Q145 pool outside of aggregates (Figure 7C–E and Appendix A; *ω*_1_ = −0.0443 s^−1^). Thus, DMD-FLIP allows for the data-based discrimination of different dynamic pools of mtHtt in living cells. Apart from its efficient denoising capacity, this must be seen as one of the major strengths of this new method.

## 4. Discussion

FLIP microscopy is widely employed to determine barriers to the diffusion of fluorescent proteins and other tagged biomolecules. When compared to FRAP, it has the important advantage of providing the cellular context, in which the diffusion, binding, and aggregation of proteins takes place. To make use of this potential, computational analysis of FLIP microscopy is of utmost importance for the proper interpretation of imaging data. In this study, a novel method is introduced, which allows for the dissection of the dynamic modes of fluorescence loss in FLIP live-cell-microscopy time-lapse data. Using simulated and experimental FLIP image sequences, it is shown that DMD-FLIP correctly identifies different dynamic contributions to the distinct fluorescence-loss signal in each compartment. DMD-FLIP employs the full range of spatiotemporal information available in the FLIP microscopy data to identify compartment-specific fluorescence-loss dynamics in the cytoplasm, nucleus, and in protein aggregates. It is, therefore, similar to coupled DMD, which has recently been successfully used in epidemiological multi-compartment systems [56]. Based on the determined time decay of fluorescence in aggregates, information about their physico-chemical properties can be inferred; while rapid fluorescence loss is found in liquid-like protein condensates, stable aggregates show very slow fluorescence decreases, due to hindered protein release. This is shown for eGFP-mtHtt, with an extended polyglutamine stretch (Figure 6 and Figure 7). At the same time, DMD-FLIP is very efficient in denoising such FLIP image data, both in space (i.e., pixel-to-pixel variation in each image) and in time (i.e., intensity profiles in each pixel position). In this regard, DMD-FLIP is as effective as state-of-the-art denoising algorithms, such as PURE-LET denoising (Figure 7). DMD-FLIP can also be employed to predict the future images of a FLIP sequence, which may not yet have been acquired. This can potentially reduce the necessary light exposure required to infer intracellular transport dynamics in experiments.

The other computational FLIP microscopy techniques we developed previously comprise the pixel-wise fitting of a stretched exponential decay function to the FLIP image sequences and the reaction–diffusion modeling of protein transport and aggregation combined with model calibration to the experimental data [11,39,40]. DMD-FLIP complements these methods in several ways; first, it is model-free, circumventing the need for model assumptions. DMD-FLIP is, therefore, simple to implement for microscopists without extensive modeling experience. Secondly, DMD-FLIP is fast compared to those methods; it takes approximately 17 s to calculate DMD modes for the image sequence in Figure 6 and Figure 7 on a zBook laptop computer and approximately 12 s on a modern desktop workstation (see Materials and Methods). Sequentially fitting a stretched exponential to each pixel (non-parallelized code) for the same image sequence on the same laptop machine takes 1862 s, i.e., more than 30 min, while it takes 290 s on a desktop workstation. Calibrating a full reaction–diffusion model for such data takes several hours on a desktop workstation [39]. Thus, DMD-FLIP is a much faster analysis method when compared to these techniques. It is also faster than PURE-Denoise.

Limitations of the current implementation of DMD-FLIP are, firstly, its inability to account properly for any abrupt lateral displacements of aggregates or other entities in the image sequence, such as those caused by Brownian motion (Figure A1). Secondly, the eigenfunctions derived by DMD are not always straightforward to associate with a particular physical process causing fluorescence loss in a given FLIP image sequence. Both limitations are inherently a result of the mathematical structure of the DMD framework; it seeks to find the best possible representation of the snapshots (i.e., images sampled in time) in a Fourier-like expansion of the eigenfunctions (Equation (8)). This does not necessarily represent the physical model which has produced the snapshots [57]. For finding this representation, a linear, discrete, time-invariant process according to Equation (3) is assumed, lying in an invariant subspace under the action of the Koopman operator, thereby identifying coherent dynamic processes in the data. The learned or inferred eigenfunctions are, therefore, not necessarily identical to the ones producing the data [42,57]. This is illustrated for the DMD reconstruction of simulated FLIP image sequences, in which the determined eigenvalues only partially resemble the analytical model (Table 1), even though the reconstruction of the data is close to perfect (Figure 2 and Figure 3). DMD is also not able to identify and describe abrupt intensity changes, such as changes in fluorescence transients or intensity oscillations or changes during stochastic particle movement, which are all examples of non-linear dynamics. The condition of linear dynamics, however, is met in FLIP image sequences, as long as no abrupt changes, such as those due to the diffusion of subcellular structures, either laterally or into and out of the focal plane of the objective, take place. In contrast, the directed movement (flow) of aggregates can be accounted for, to some degree, as shown in the simulations of Appendix C, and the experimental sequence in Appendix F (see also Appendix A). This is also expected, as directed motion is highly correlated between frames, thereby resembling a coherent flow pattern, for which DMD was originally developed [41]. Increasing the rank of the DMD approximation of snapshots improved the reconstruction quality only slightly in those cases, but the stochastic component of the dynamics was still not properly described (not shown). It has to be emphasized, though, that neither pixel-based bleaching analysis nor our reaction–diffusion modeling of FLIP data can account for the sudden movement of subcellular structures [11,39,40].

## 5. Conclusions and Outlook

This study presents an easy-to-use computational method for the analysis of time-lapse FLIP imaging of living cells. DMD-FLIP accurately decomposes fluorescence-loss dynamics in various subcellular compartments, thereby allowing for the rapid and straightforward detection of barriers and obstacles to protein diffusion without the need for demanding modeling approaches. DMD-FLIP allows microscopists to efficiently denoise their FLIP-image data and to quickly obtain spatiotemporal maps of fluorescence-loss dynamics. The ability of DMD-FLIP to efficiently decompose the fluorescence-loss kinetics of different subcellular regions into distinct dynamic modes enables the study of protein dynamics at each time scale individually. This novel DMD-FLIP method is suitable for denoising spatiotemporal FLIP data for better visualization and further analysis. The reconstruction of partial FLIP data with incomplete fluorescence loss can enable optimization of FLIP experiments with minimal light exposure to reduce cell stress and prevent artefacts. DMD-FLIP also allows for the development of reduced models of intracellular dynamics and should thereby set the stage for the comprehensive modeling of protein transport and aggregation in living cells.

Several extensions of classical DMD can be considered to overcome the current limitations of DMD-FLIP in the future, which primarily include its inability to properly describe the abrupt translational motion of fluorescent entities and the accurate, but non-physical, description of kinetic data. Firstly, multiresolution DMD could be implemented, which allows for the improved description of dynamic modes at multiple and distinct time scales [44]. Secondly, physics-informed DMD should be employed to restrict the possible space of solutions to the minimization problem in Equation (4) to a subspace based on prior knowledge about the physical system under investigation [58]. This leads to a reformulation of the optimization task in DMD to a Procrustes problem, which could, for example, be used to model the convection–diffusion of aggregates or other subcellular structures in FLIP data via a tri-diagonal constraint to the admittable reconstruction of matrix *A* [58]. Another possibility to improve the analysis of FLIP, and other dynamic microscopy data, is to approximate the linear, but infinite, Koopman operator using non-linear measurement functions of the state variable x, as performed in extended dynamic mode decomposition [59]. This would potentially also allow for the description of a combination of stochastic and deterministic dynamics, as is often encountered in live-cell time-lapse microscopy. The ultimate goal of data-driven modeling of microscopy data is to infer the real underlying equations governing the experimentally observed dynamics. This has been attempted by calibrating multi-compartment, reaction–diffusion systems in our previous studies [7,39,40]. A very attractive strategy would be to attempt to discover and parametrize such equations directly from the data, including an account for protein diffusion using sparse identification strategies of the underlying partial differential equations [60]. This will be the focus of future research.

## Figures and Tables

**Figure 1 sensors-22-04731-f001:**
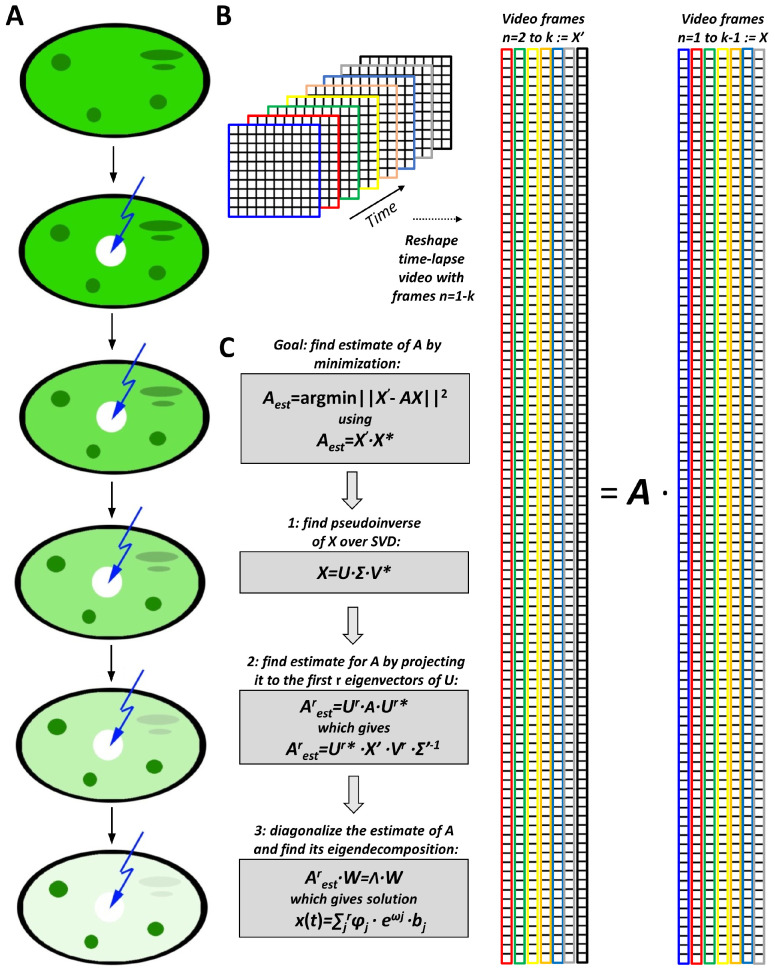
Principle of DMD-FLIP microscopy. Fluorescence loss in photobleaching (FLIP) is based on the repeated bleaching of a small subcellular area and monitoring fluorescence loss in the entire cell by intervening confocal image acquisition (**A**). For subsequent DMD analysis, the FLIP time-lapse image sequence is reshaped into two matrices, in which each image becomes a column (**B**). The first matrix, X, runs from the first image (*n* = 1, blue) to the next last image (*n* = *k* − 1, grey) stacked as columns. The second matrix, X′, runs from the second image (*n* = 2, red) to the last image of the video (*n* = *k*, black). The time-evolution of the video sequence is modeled as a matrix equation of the form X′ = A∙X, and the aim of DMD is to find the best possible approximation to this matrix (**C**). A singular-value decomposition of X allows for a rank-r approximation with the most dominant singular values giving the rank-reduced matrix, X^r^. The pseudoinverse of this matrix can be computed, which allows for deriving an estimate of the matrix A, which is A^r^_est_. Spectral decomposition of this matrix provides its eigenvectors as columns of the matrix, W, and corresponding eigenvalues organized in the matrix, Λ. A linear combination of the derived dynamic modes, φ_j_, mode amplitudes, b_j_, and eigenvalues, ω*_j_* = log(λ_j_)/Δt, allows for reconstructing the full dynamic information of the original image data. Note that ‘*’ indicates the conjugate transpose of a matrix.

**Figure 2 sensors-22-04731-f002:**
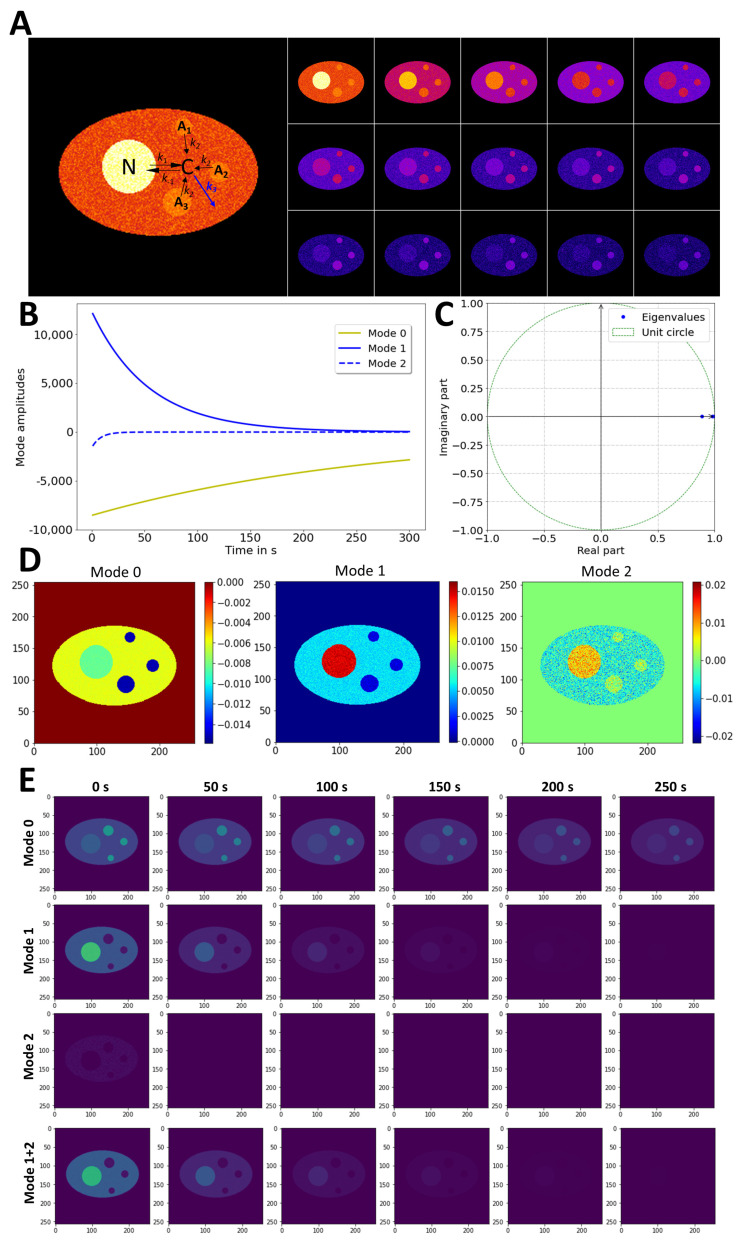
DMD of synthetic FLIP image series with cytoplasmic aggregates. Simulated FLIP images with nucleocytoplasmic exchange (rate constants, *k*_1_ and *k*_−1_) and protein release from aggregates (rate constant *k*_2_) as well as bleaching in the cytoplasm with rate constant *k*_3_ (blue arrow, (**A**)). Montage of simulation with every 20th frame shown in left panel of (**A**,**B**), mode decays obtained by DMD of rank 3. (**C**) Eigenvalues plotted on unit circle. (**D**) The 2D maps of identified dynamic modes. (**E**) Reconstructed time evolution of each identified mode (upper three panels) and sum of mode 1 and 2 (lowest panel). See text for further explanations.

**Figure 3 sensors-22-04731-f003:**
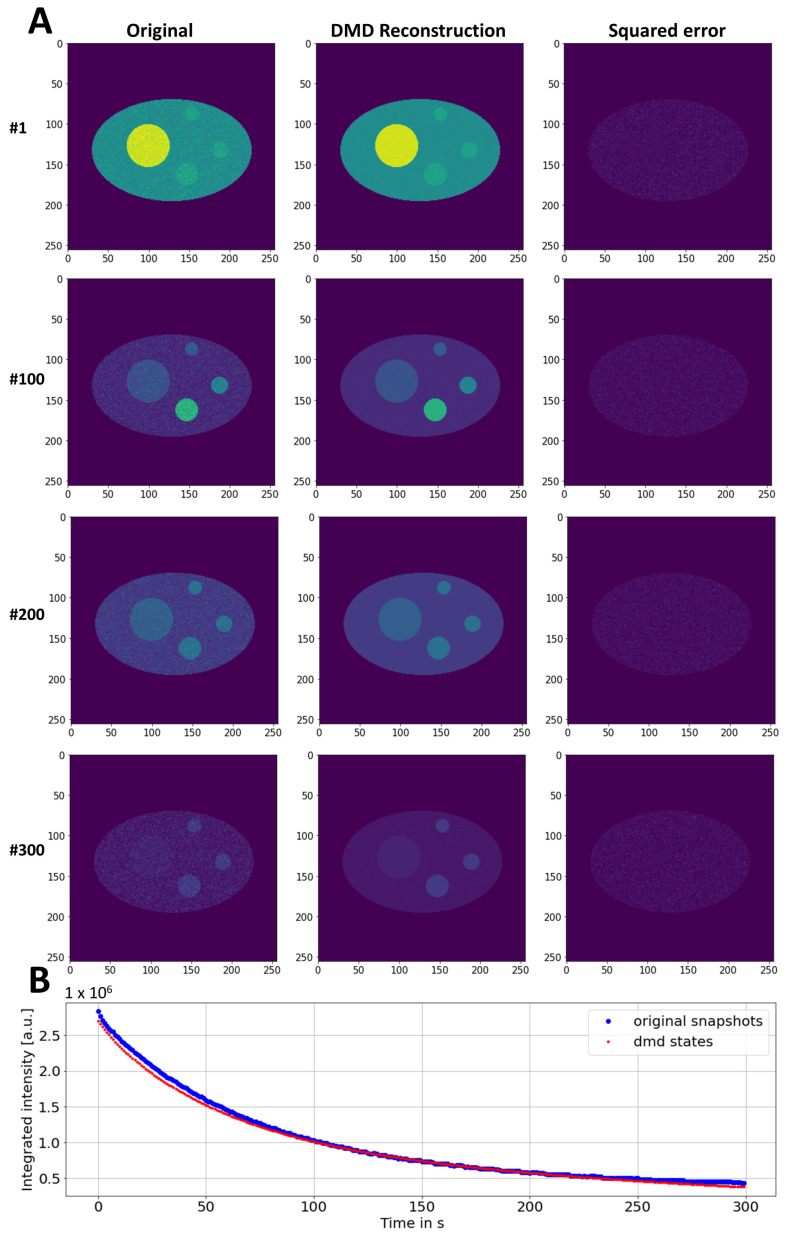
DMD reconstruction of synthetic FLIP images with decay in aggregates. (**A**) Comparison of simulated and DMD-reconstructed FLIP image series. The squared error with identical intensity scaling is shown in the right column. Legend at the left margin shows frame number (‘#’). (**B**) Integrated intensity of simulated (blue dots) and reconstructed image series (red dots). See text for further explanations.

**Figure 4 sensors-22-04731-f004:**
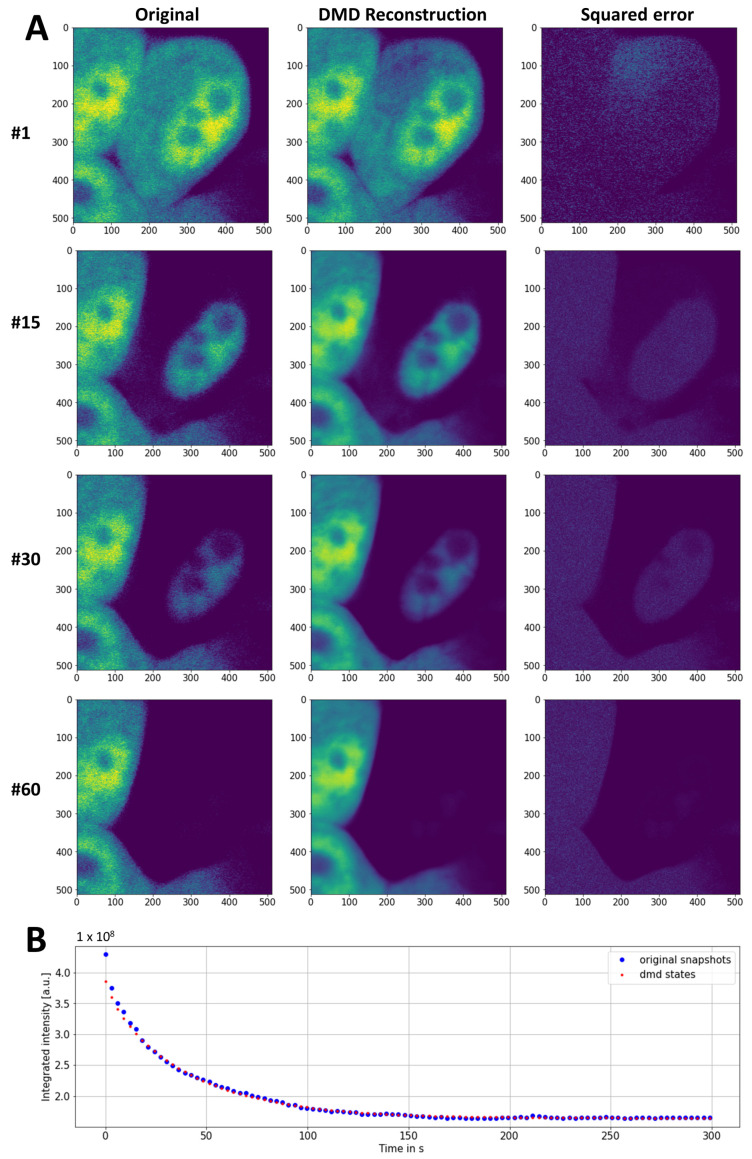
DMD reconstruction of experimental FLIP images of eGFP in McA cells. (**A**) FLIP experiment was carried out on a confocal microscope using McA cells expressing eGFP. Bleaching was performed in the cytoplasm every 3 sec, with image acquisition in between (see white circle in upper left panel for laser spot position in (**A**)). (**A**) Comparison of experimental and DMD-reconstructed FLIP image series. The squared error with identical intensity scaling is shown in the right column. Legend at the left margin shows frame number (‘#’). (**B**) Integrated intensity of simulated (blue dots) and reconstructed image series (red dots). See text for further explanations.

**Figure 5 sensors-22-04731-f005:**
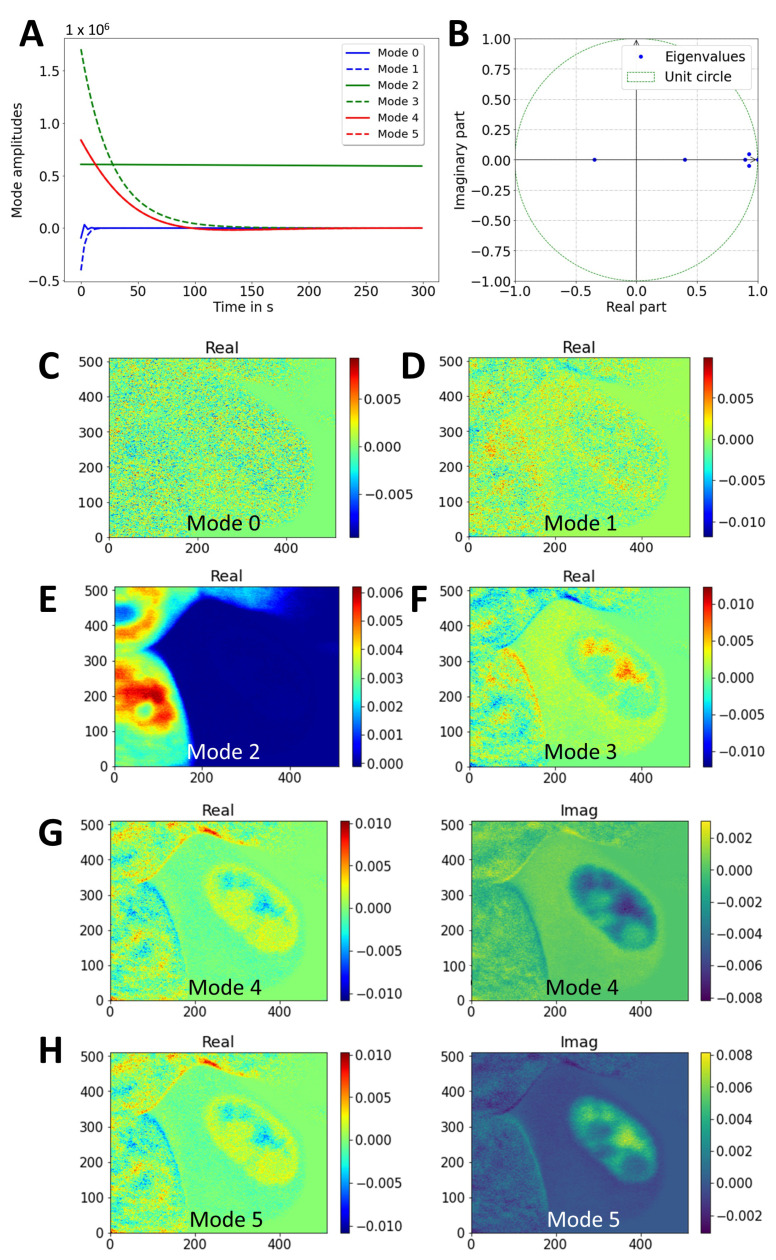
Analysis of DMD of FLIP image sequence of eGFP expressed in McA cells. (**A**) Mode decays obtained by DMD of experimental sequence from Figure 4 with rank 3. (**B**) Eigenvalues plotted on unit circle. (**C**–**F**) The 2D maps of the identified dynamic modes, i.e., real part of mode 0 (**C**), mode 1 (**D**), mode 2 (**E**), and mode 3 (**F**). All imaginary parts of those modes were zero (not shown). (**G**,**H**) Real and imaginary part of modes 4 (**G**) and 5 (**H**). Note that the mode images are flipped vertically compared to the original sequence (compare Figure 4). See text for further details.

**Figure 6 sensors-22-04731-f006:**
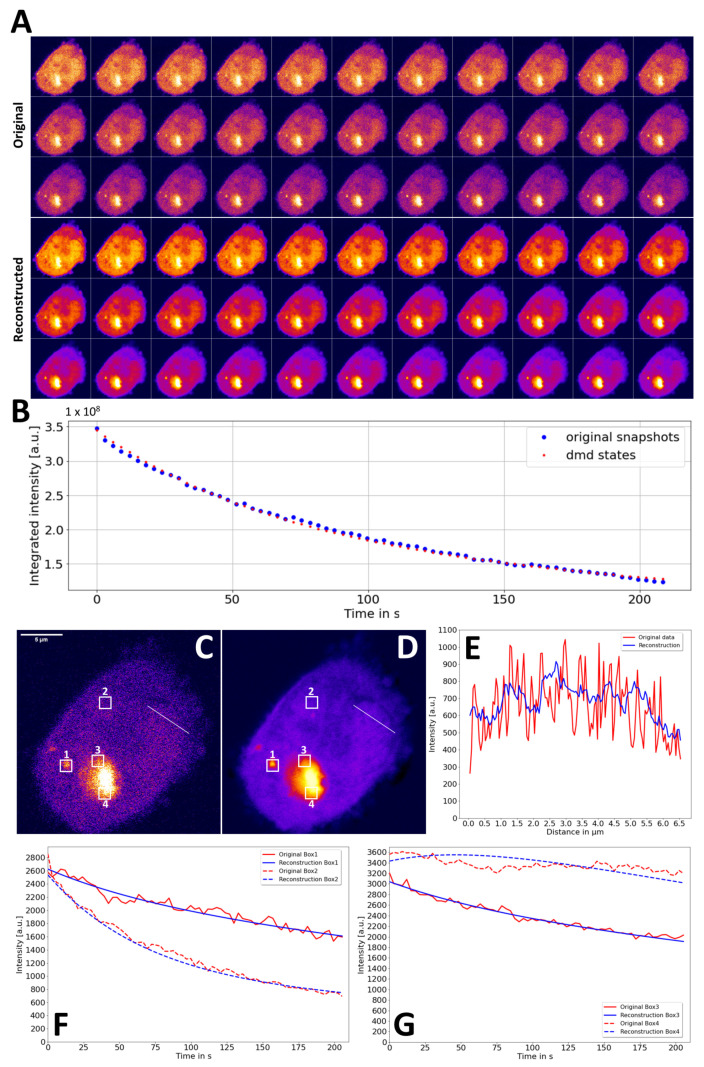
DMD reconstruction of experimental FLIP images of eGFP-Q145 expressed in CHO cells. A FLIP experiment was carried out on a confocal microscope using CHO cells expressing eGFP-Q145. Bleaching was performed in the cytoplasm every 3 s, with image acquisition in between (see white circle in upper left panel for laser spot position in (**A**)). (**A**) Comparison of original and DMD-reconstructed FLIP image series (every 2nd frame, i.e., every 6 s is shown). (**B**) Integrated intensity of experimental (blue dots) and reconstructed image series (red dots). (**C**,**D**) Last frame of original (**C**) and DMD-reconstructed (**D**) image stack with line and boxes indicated. E–G Line profile (**E**) along line in (**C**,**D**) and intensity time decay (**F**,**G**) for boxes indicated in (**C**,**D**) are shown. Original (red lines) and DMD-reconstructed (blue lines).

**Figure 7 sensors-22-04731-f007:**
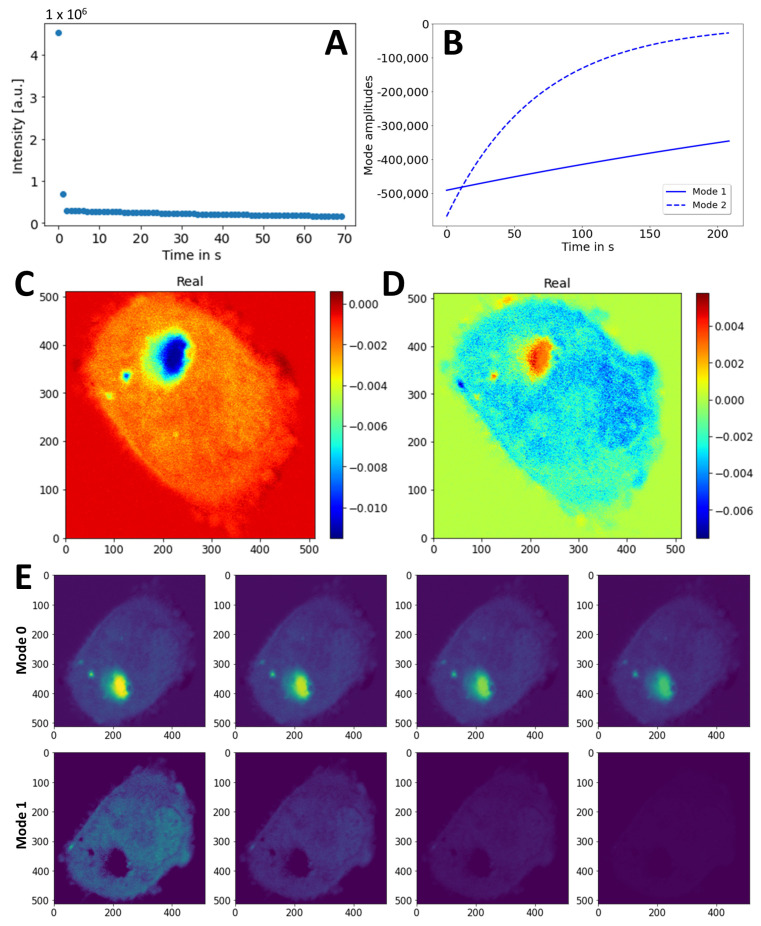
Analysis of DMD of FLIP image sequence of eGFP-Q145 expressed in CHO cells. (**A**) Singular values calculated from reshaped image stack. (**B**) Mode decays obtained by DMD of experimental sequence from Figure 6 with rank 2. The 2D maps of real part of identified dynamic modes, with mode 0 (**C**), mode 1 (**D**). Imaginary parts were zero (not shown). (**E**) Reconstruction of mode 0, upper panel, and mode 1; lower panel with identical intensity scaling. Every 20th frame is shown; see text for further details.

**Table 1 sensors-22-04731-t001:** Comparison of eigenvalues of FLIP simulation with estimated eigenvalues of DMD and pixel-wise exponential fitting.

Eigenvalues *	FLIP Simulation	DMD Reconstruction	Pixel-Wise Fitting ^#^
$-k_{2}$	*l*_1_ = −0.0050 s^−1^	ω_0_ = −0.0038 s^−1^	*k*_agg._ = −0.0051 s^−1^
$-(2k_{1}+k_{3}+\sqrt{k_{3}^{2}+4k_{1}\cdot k_{-1}})$	*l*_2_ = −0.6342 s^−1^	ω_2_ = −0.1162 s^−1^	-
$-(2k_{1}+k_{3}-\sqrt{k_{3}^{2}+4k_{1}\cdot k_{-1}})$	*l*_3_ = −0.0158 s^−1^	ω_1_ = −0.0189 s^−1^	*k*_cell._ = −0.0139 s^−1^

* Theoretical expression based on analytical FLIP model in Appendix B, Equations (A7)–(A9). ^#^ Determined using a stretched exponential function in PixBleach [11]. Rate constants were determined as inverse of the mean time constant values in the aggregates (*k*_agg_) or the inverse of the averaged time constant values in the nucleus and cytoplasm (*k*_cell_). The stretching parameter was *h* = 0.999, *h* = 1.028, and *h* = 1.078 in aggregates, nucleus, and cytoplasm, respectively.

## Data Availability

All data analyzed during this study are included in this published article [and its Appendix A]. The original datasets generated and/or analyzed during the current study are available in the GITHUB repository, DanielW-alt/DMD-FLIP (github.com).

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
