# Peer review of "Dynamic Mode Decomposition of Fluorescence Loss in Photobleaching Microscopy Data for Model-Free Analysis of Protein Transport and Aggregation in Living Cells"

_sensors, 2022, doi:10.3390/s22134731_

Round 1

Reviewer 1 Report

Wüstner reports an analysis method of fluorescence loss in photobleaching (FLIP) live-cell image sequences based on dynamic mode decomposition (DMD). The author claims that this method allows model-free detection of barriers to protein diffusion and denoising confocal time-series image data.

Although the DMD-FLIP described in this manuscript might have high applicability to live-cell protein aggregation studies, there are major concerns to be clarified and some minor points to be improved. It is not easy to understand the advantages of DMD-FLIP by the present manuscript. The present manuscript lacks comparison that demonstrates the advantages in a way to interest readers. Therefore, I recommend the manuscript be published in Sensors after revisions to address the following criticisms.

Major points:

  1. The author presents the diffusion and rate constants obtained for nucleo-cytoplasmic exchange of mutant Huntingtin proteins based on DMD-FLIP. However, it is hard for readers to find the constants in the main text. In addition, it is unclear how DMD-FLIP is quantitively advantageous over other FLIP analysis methods. Therefore, please summarize the constants in a table(s) and compare them with those obtained by other FLIP analysis methods. Finally, please note that the author’s group published a paper [22] in which FLIP data for the same protein molecules were analyzed by a different method.

  1. Please explain whether the obtained rate constants are physiologically relevant.

  1. The author assumes single rate constants for k1, k-1, and k3 in differential equations (L667-669). Please describe whether the DMD-FLIP can analyze with multiple rate constants for k1, k-1, and k3. If it is possible, the rate constants determination may be more precise.

Minor points:

  1. What does “FCS” stand for? (L142 and 145).

  1. “2” of “MgCl2” and “CaCl2” should be subscripted (L151-152)

  1. Please change the expression “destroy the fluorescence” to a more appropriate expression (L164).

  1. “to improve” may be “to improve” (L579).

  1. “Here, the analytical FLIP model used in simulations described in Fig. 2, 3 and Appendix A is derived.” may be “Here, the analytical FLIP model used in simulations described in Fig. 2, 3 is derived.”(L651).

  1. “compare Fig. 1” may be “compared to Fig. 1” (L717).

Reviewer 2 Report

The paper is well written and well organized. The author presents a computational method for analysis of fluorescence loss in photobleaching imaging time-series of living cells. The author shows that this novel model-free method resolves complex spatio-temporal fluoresence loss kinetics through the exploitation of a dynamic mode decomposition of fluorescence loss in photobleaching live-cell iage sequences. The method is applied on aggregation of proteins, a particularly critical problem in neurodegenerative diseases. In addition, the method should be suitable for denoising spatio-temporal fluorescence data for better visualization. In my opinion, the paper will have potentially a large impact for a wide range of scientists and physicians, among others, biologists, neuro-scientists, physicists, and biochemists.

Author Response

Response: I thank the reviewer for the positive assessment of the manuscript and for appreciating its potential impact on future studies of neurodegenerative diseases.

Reviewer 3 Report

Dear Author,

Before considering the publication of the manuscript, some revisions should or could be done.

The introduction could be shortened.

The figures should be improved. A higher text font size should be used for axes labels (for the Figures shown on pages: 13, 14, 15, 23 and 24).

The graph shown on Figure 5, point A, is incomplete, the Mode 5 is not shown on this graph.

Best regards

Round 2

Reviewer 1 Report

In the revised manuscript, the author made efforts to clarify the ambiguities that I pointed out and addressed the questions mostly.

The revision facilitates understanding of the advantages of DMD-FLIP.

The manuscript is of publication quality, pending some minor English errors as described below.

Please carefully check the English language over the entire manuscript.

1. In the Materials and Methods section, please insert a space before a unit (e.g. 1mM) and between words (5mMglucose).

2. I am wondering if “physical mechanism” may be “physiological mechanism” (L343).

3. “Tab. 1” may be “Table 1” (L535). I saw this abbreviation for the first time. Is the abbreviation commonly used?

4. “Fig. 2, 3” may be ”Figs. 2, 3.” I am sorry that I overlooked this point in the first-round review.

Author Response

1. In the Materials and Methods section, please insert a space before a unit (e.g. 1mM) and between words (5mMglucose).

Response: this has been corrected in the revised manuscript.

2. I am wondering if “physical mechanism” may be “physiological mechanism” (L343).

Response: this has been changed to 'biophysical mechanism' in the revised manuscript. 'Physiological' would have been less adequate.

3. “Tab. 1” may be “Table 1” (L535). I saw this abbreviation for the first time. Is the abbreviation commonly used?

Response: the abbreviation has been removed, and it says 'Table 1' in the revised manuscript.

4. “Fig. 2, 3” may be ”Figs. 2, 3.” I am sorry that I overlooked this point in the first-round review.

Response: this has been corrected in the revised manuscript.